# The Inflammatory Profile of Obesity and the Role on Pulmonary Bacterial and Viral Infections

**DOI:** 10.3390/ijms22073456

**Published:** 2021-03-26

**Authors:** Franziska Hornung, Julia Rogal, Peter Loskill, Bettina Löffler, Stefanie Deinhardt-Emmer

**Affiliations:** 1Institute of Medical Microbiology, Jena University Hospital, Am Klinikum 1, D-07747 Jena, Germany; franziska.hornung@med.uni-jena.de (F.H.); bettina.loeffler@med.uni-jena.de (B.L.); 2Department of Women’s Health, Research Institute for Women’s Health, Eberhard Karls University, Calwerstrasse 7, 72076 Tübingen, Germany; julia.rogal@igb.fraunhofer.de (J.R.); peter.loskill@uni-tuebingen.de (P.L.); 3Fraunhofer Institute for Interfacial Engineering and Biotechnology IGB, Nobelstrasse 12, 70569 Stuttgart, Germany; 4Center for Sepsis Control and Care, Jena University Hospital, D-07747 Jena, Germany

**Keywords:** obesity, lung infection, adipocytokines, bacteria, viruses, obesity paradox

## Abstract

Obesity is a globally increasing health problem, entailing diverse comorbidities such as infectious diseases. An obese weight status has marked effects on lung function that can be attributed to mechanical dysfunctions. Moreover, the alterations of adipocyte-derived signal mediators strongly influence the regulation of inflammation, resulting in chronic low-grade inflammation. Our review summarizes the known effects regarding pulmonary bacterial and viral infections. For this, we discuss model systems that allow mechanistic investigation of the interplay between obesity and lung infections. Overall, obesity gives rise to a higher susceptibility to infectious pathogens, but the pathogenetic process is not clearly defined. Whereas, viral infections often show a more severe course in obese patients, the same patients seem to have a survival benefit during bacterial infections. In particular, we summarize the main mechanical impairments in the pulmonary tract caused by obesity. Moreover, we outline the main secretory changes within the expanded adipose tissue mass, resulting in chronic low-grade inflammation. Finally, we connect these altered host factors to the influence of obesity on the development of lung infection by summarizing observations from clinical and experimental data.

## 1. Introduction

Obesity is becoming a growing health problem worldwide. It is generally defined as a condition of increased adipose tissue mass [1] and can be further specified as an accumulation of body mass beyond physical requirements [2]. The WHO describes obesity and overweight as excessive fat accumulation leading to higher morbidity rates for various health problems [3]. In 1842, Adolphe Quetelet conducted pioneering work in analyzing the differences in the weight status of individuals [4]. The Quetelet dindex is calculated by dividing weight by height squared and is known nowadays as the Body Mass Index (BMI), which is still the most common way of classifying obesity [5]. Using this scale, overweight is defined as a BMI greater than or equal to 25 and obesity as a BMI greater than or equal to 30 [3].

With more than 1.9 billion overweight adults in 2016 and a worldwide tripling of the number of obese persons since 1975 [3], obesity has spread around the world and now affects a considerable part of the human population. This high prevalence represents a huge problem for our health care systems, because “Corpulence is not only a disease itself but the harbinger of others”, as Hippocrates already knew more than two millennia ago [6]. Metabolic diseases (e.g., type 2 diabetes mellitus), bone and soft tissue pathologies (e.g., osteoarthritis), and cardiovascular diseases (e.g., hypertension) belong to the main comorbidities of obesity. Moreover, obesity also leads to impaired lung function, increased occurrence of asthma, and obstructive sleep apnoea syndrome (OSA) [7]. Since obesity contributes to a wide variety of comorbidities, an excess of adipose tissue mass is expected to lead to a variety of molecular changes in the body. Regarding respiratory infections, an increased sensitivity has been observed, which might be connected to the above-mentioned increased incidence of comorbidities but also by a chronic low-grade inflammatory status.

In the process of infection progression, obese patients have been reported to show benefits or disadvantages compared to normal-weight subjects depending on the type of infection. On the one hand, the “obesity paradox” describes a benefit for the obese, e.g., in sepsis [8]. On the other hand, certain infections take a markedly more severe course in obese patients compared to normal-weight patients, as the current coronavirus disease (COVID-19) pandemic clearly illustrates [9,10]. Thus, the “obesity paradox” proposes that obese patients, although presenting numerous comorbidities, show a survival benefit [11].

Here, we summarize the main mechanical impairments in the pulmonary tract caused by obesity and particularly the molecular alterations within adipose tissue that go along with its expansion. Most notably, we outline the main secretory changes within the expanded adipose tissue mass, resulting in the establishment of chronic low-grade inflammation. To mimic obesity in different experimental settings either in vivo or in vitro specialized models are needed to study the underlying mechanisms. Our review additionally focuses on model systems suitable for mechanistic studies of biology and signal transduction in adipose tissue. With this knowledge, we connect the mentioned altered host factors to the influence of obesity on the development of lung infection by summarizing observations from clinical as well as from experimental data. Finally, we discuss the differences between bacterial and viral infections concerning the effect of obesity.

## 2. The Mechanic Pulmonary Complications of Obesity

An increase in adipose tissue and fat deposition throughout the body has a direct impact on the upper airway and pulmonary function. Especially, the deposition of fat in the upper part of the body reduces the airway lumen and impairs the muscles important in maintaining the full respiratory function, resulting in OSA [12].

The combination of severe obesity and hypoventilation is described as Obesity-Hypoventilation syndrome (OHS) [13]. According to Sharp et al. [14], the chest wall and total respiratory compliance for obese patients range from 92% to 80% and for patients with additional OHS from 44% to 37% to of normal values. A possible reason for this deficit could be an expanded blood volume and the subsequent closure of corresponding air passages, two parameters influencing lung compliance. Furthermore, the resistances of the airway, chest wall, and respiratory system are elevated in obese individuals, especially in OHS patients, and might be caused by a reduction in lung volume [14]. These changes are also associated with an impaired Work of Breathing (WOB). Simply and severely obese subpatients have a 70% to 280% higher WOB than normal and accordingly an extremely increased cost of breathing [15]. The work of breathing is defined as the energy, which is needed to inhale and exhale breathing gas, referred to as work per unit volume. Moreover, simple obese and OHS patients show a decrease in strength and endurance of the inspiratory and expiratory muscles, possibly due to enlargement or overstretching of the diaphragm [16]. The lung volumes of simple obese patients are generally not profoundly altered, except for a reduction in the expiratory reserve volume, potentially caused by the displacement of the diaphragm into the chest. Only very severe types with additional OHS show additional alterations in other lung volumes [7]. This effect is most likely not caused by a simple BMI increase but by a change in fat distribution, e.g., in the chest wall, abdominal wall, or abdomen [7]. Especially eucapnic morbidly obese patients show a difference in the pattern of breathing because their respiratory rate at rest is around 40% higher than normal [17]. Obese patients, especially those with small lung volume and in a supine position, are often hypoxemic, leading to a mismatch of ventilation and perfusion, which influences the gas exchange in the lung [18].

Even though obese individuals have an almost 25% higher oxygen consumption at rest than normal, young individuals show similar performance during exercise in terms of heart rate and blood pressure [17]. Merely at resting-phases between exercises, higher respiratory rate (RR) and lower tidal ventilation values can be observed [19]. Obesity can cause a number of further medical complications concerning the pulmonary system: For example, the risk of developing aspiration pneumonia is increased in obese patients, mainly caused by a higher volume of gastric fluid, lower pH in the gastric system, elevated pressure in the abdomen, or gastroesophageal reflux [20]. Obese subjects are at higher risk for postoperative thromboembolic disease [21] and pulmonary embolism [22] resulting in challenging anesthesia in obese patients [23]. Difficulties can, for instance, arise during intubation with atelectasis being a severe side effect in 30% of obese patients [23]. Furthermore, difficulties in mechanical ventilation and acute postoperative respiratory events can occur [24]. These impairments could represent a physical basis for the increased susceptibility to infections of the pulmonary tract and additional hinder its treatment.

## 3. The Inflammatory Activity of Adipose Tissue

Adipose tissue is classified into two main types, white adipose tissue (WAT) and brown adipose tissue (BAT). WAT is the more predominant form in the human body and plays a major role in energy storage. Thermogenesis is the main function of BAT. It was thought to be rather present in small mammals and human neonates [25]. Nedergaard et al. indeed revealed its presence in adult tissue [26].

Adipocytes are in general the main cell type within adipose tissue. They can be further classified according to their microscopical appearance [27]. A unilocular positioned lipid-vacuole characterizes white adipocytes, the predominant form in the WAT. Brown appearing adipocytes are defined by multilocular lipid-vacuoles and an increased amount of mitochondria, which is connected to their function in heat production in BAT. Both, WAT and BAT, are capable to transdifferentiate between the subtypes in response to the physiological conditions [28]. The intermediate cell forms are called beige adipocytes [29]. White adipocytes are additionally able to transdifferentiate into epithelial, milk-producing cells in the breast of pregnant women. These cells are named after their visual appearance, the pink adipocytes [30].

Besides adipocytes, also pre-adipocytes, endothelial cells, fibroblasts, leukocytes, and bone-marrow-derived macrophages are part of adipose tissue [31]. The number of macrophages positively correlates with body mass, adipocyte size, and expression of pro-inflammatory cytokines [32].

One of the main physiological functions of WAT is the regulation of fat reservoirs in the body via triacylglycerols (TG) stored in adipocytes. The mobilization and storage of TG must be well balanced with and connected to the energy intake and expenditure of the whole body. In this context, the lipolysis, i.e., the metabolization of TG, is also regulated by the autonomic nervous system, in particular adrenergic and cholinergic neurons [33].

In addition to energy storage, adipose tissue has an important endocrine function secreting a number of crucial soluble factors: Specific to adipose tissue are the so-called “adipocytokines” or “adipokines”, including, e.g., adiponectin, leptin, resistin, and visfatin [31]; described in more details in the next chapter. Other important produced factors include the cytokines tumor necrosis factor (TNF), interleukin-6 (IL-6), interleukin-1 (IL-1), CC-chemokine ligand 2 (CCL2), plasminogen activator inhibitor type I (PAI-I), and a number of complement factors [34,35]. Most of these factors are known as pro-inflammatory mediators that induce immune cell infiltration and play a major role in the development of infectious diseases.

According to the location of the deposition, adipose tissue can be further classified in subcutaneous adipose tissue (SCAT) and visceral adipose tissue (VAT). The excessive production of fat tissue during weight gain leads to a depletion of storage capacities and can result in ectopic lipid accumulation in visceral body cavities, skeletal muscle, or liver tissue [36,37]. This phenomenon can for instance play a major role in the development of insulin resistance [38].

An increase in adipose tissue mass during weight gain can be on a cellular level either orchestrated by an increase in the size of the adipocytes (hypertrophy) or their number (hyperplasia). Besides simple mass expansion, the tissue additionally passes through a process of remodeling characterized by extracellular matrix (ECM) overproduction, increased immune cell infiltration, and higher pro-inflammatory response [39].

Specifically, the crosstalk between adipocytes and macrophages plays an important role in the remodeling [40]. The elevated infiltration leads to the expression of macrophage-related inflammatory genes [41]. In this context, Sun et al. [39] proposed four different mechanisms as potential key initiator processes of macrophage migration to the adipose tissue: adipocyte death, chemotactic regulation, hypoxia, and fatty acid flux. Thus, the excess of body weight via increased adipose tissue mass results in a state of low-grade chronic inflammation (Figure 1).

Saltiel et al. [42] suggested the occurrence of a “phenotypic switch” during polarization of macrophages from an anti-inflammatory M2 type to the M1 form. In lean individuals, the polarized M2 type seems to encourage the normal function of adipocytes by promoting repair of tissue and angiogenesis, i.e., the sprouting of new vessels to ensure an appropriate oxygen supply. In obese individuals, the M1 type is fostering an inflammatory milieu in the adipose tissue by secreting factors like TNF-α, inducible nitric oxide synthase (iNOS), C-C chemokine receptor type 2 (CCR2) or monocyte chemotactic protein 1 (MCP1). Sun et al. [40] proposed that this switch in polarized macrophages defines the fate of adipocyte function and the overall inflammatory profile of the tissue. They also discriminated between healthy and pathological adipose tissue expansion. In the former, an increase in mass happens mainly via a growing number of small adipocytes, recruitment of other stromal cells, maintenance of oxygen supply, and only minimal induction of ECM production and inflammation. In contrast, pathological expansion is defined as a rapid increase in the size of existing adipocytes with hypoxia due to decreased blood vessel formation, massive ECM deposition, and a higher number of macrophages, especially the pro-inflammatory M1 type, leading to a state of chronic inflammation.

In summary, this chapter highlights the complex composition and inflammatory potential of adipose tissue in obese patients. Hence, it is assumed that the calculation of a person’s BMI by using solely two parameters might lead to misclassifications and excludes important parameters such as percentage body fat (PBF) or rather molecular biomarkers. With regard to the investigations of predispositions for various diseases, it should be considered to include more parameters. In this context, DeLorenzo et al. [43] summarized four obese phenotypes: (1) normal weight obese, (2) metabolically obese normal weight; (3) metabolically healthy obese and (4) metabolically unhealthy obese. Here additional factors concerning body fat composition and distribution (such as fat mass, glucose-levels or CRP-levels) are taken into account. This would also specify the analysis of the impact of obesity on infectious diseases.

## 4. Adipocytokines Produced by Adipose Tissue

Since the discovery of leptin in 1994, adipose tissue is also known as an endocrine organ, besides its function contributing to energy storage [44]. It can express and secrete a range of proteins, which are termed adipocytokines or adipokines, because of their main but not exclusive production in this specialized type of tissue [31]. In the following section, the main adipokines and their function in inflammatory processes are described. Additionally, we highlight their origin and detected alterations during weight gain (Figure 1).

### 4.1. Pro-Inflammatory Molecules of the Adipose Tissue

Leptin is the best-characterized adipokine, mainly produced by adipocytes. It was discovered in the 1960s with the help of parabiotic experiments with two mouse strains, the *ob/ob* (obese) and *db/db* (diabetes) mutant. This procedure aims at the surgical joining of two mice, leading to the coupling of their blood circulation and enabling the analysis of different physiological or hormonal processes [45]. Both of the investigated mutants result in an obese phenotype. By conjoining individuals of those strains with each other or with lean mice, respectively, different weight changes were observed. Remarkably, the *db/db* mouse overexpressed a factor but was not able to respond to it with weight loss. In fact, the *ob/ob* mouse could not produce this factor but reacted to it by losing weight while connected with the *db/db* mutant or the respective wildtype [46]. 40 years later, this factor and product of the *ob* gene that seemed to orchestrate the bodyweight of the two mutant strains was named leptin (Greek: leptos = thin) [44]. The gene product of *db* was termed ObR, the corresponding receptor [47].

An overview of the main downstream branches of leptin are schematically displayed in Figure 2. Leptin is a 16 kDa polypeptide that shows structural similarities to the long-chain cytokines IL-6, IL-12, or G-CSF, all known to contribute to inflammation [48]. The leptin-receptor is a type I cytokine receptor type and exists as six alternatively spliced isoforms with varying intracellular, cytoplasmic parts [49]. One long (ObRb), four short (ObRa, ObRc, ObRd, and ObRf) and one soluble form (ObRe) exist. Only the long isoform possesses the entire intracellular domain, with the conserved tyrosine residues (Y985, Y1077, Y118) [50]. Besides the expression in the hypothalamic region of the brain, where the main function of appetite regulation is orchestrated via leptin signaling, other tissues also express leptin receptor isoforms, including the heart, placenta, liver, muscle, kidney, pancreas, spleen thymus, prostate, testes, ovary, small intestine, and colon [51]. Remarkably, Tsuchiya et al. [52] revealed that the human leptin receptor also occurs in lung tissue.

Leptin exerts its function in the hypothalamus via the activation of anorexigenic POMC (Proopiomelanocortin) neurons and orexigenic NPY (Neuropeptide Y)/AgRP (Agouti-related Protein) neurons [53]. The most prominent downstream signaling component of the leptin receptor is the JAK (Janus kinase)/STAT (signal transducer and activator of transcription) pathway [54]. An important downstream gene is SOCS3 (Suppressor of cytokine signaling 3), which itself acts as a potent negative regulator of the leptin signaling, by inhibiting JAK2 [55]. Another signaling branch activated via leptin is the MAPK (Mitogen-activated protein kinase) pathway. Here, the SH2 domain of the phosphatase SH2- containing protein tyrosine phosphatase 2 (SHP2) binds to pY985, becomes phosphorylated by JAKs, which finally activates MAPK extracellular signal-related kinase (ERK1/2) via recruitment of growth factor receptor-bound protein 2 (Grb2) [56]. Leptin also activates Akt by triggering the phosphatidylinositol 3 kinase (PI3K) signaling pathway. Akt inhibits forkhead box O1 (Foxo1), the mediator of the previously mentioned function of leptin in the hypothalamus [57]. Furthermore, downstream of Akt is the Ser/Thr kinase mTOR (mammalian target of rapamycin), a mammalian sensor for the availability of nutrients and stimulator of cell growth, protein biosynthesis, and proliferation [58]. Leptin stimulates the AMPK (adenosine monophosphate-activated protein kinase) pathway, but the outcome differs between the tissues. While activated in hepatocytes and muscle [59], AMPK is inhibited in the hypothalamus, resulting in inhibition of food intake [60].

The observations gained from the parabiosis experiments reflect one main function of leptin: the regulation of food intake and energy expenditure via the leptin-hypothalamus axis. In the hypothalamus, leptin activates anorexigenic neurons, thereby decreasing food intake; at the same time, it leads to the inactivation of orexigenic neurons that stimulate appetite and intake [53]. Counterintuitive to this role in food intake regulation, the level of leptin in the systemic circulation of obese persons is elevated and positively correlated with adipose mass. This hyperleptinemia is a consequence of developing leptin resistance [61]. Several potential explanations for this effect are discussed. On the one hand, the action of a negative feedback loop orchestrated by SOCS3, activated by leptin could serve as a potential link to the leptin resistance. The protein tyrosine phosphatase 1 B (PTP1B) is capable to inhibit the leptin receptor activity by dephosphoryltion of JAK2 [62]. On the other hand, disturbances in the hypothalamic neuronal wiring, impairments in the transport of leptin to the brain, the ObR trafficking, ER stress, or inflammation itself are discussed as potential key events for the development of leptin resistance [49].

Besides its regulatory function in the hypothalamus, leptin itself is defined as a pro-inflammatory adipokine and plays a major role in innate and adaptive immunity [63]. In monocytes, it induces increased production of TNF, IL-6, and ROS as well as cell proliferation [64], thereby stabilizing their activation, phagocytotic activity, and cytokine production [25]. In turn, leptin expression is also elevated by pro-inflammatory cytokines such as TNF and IL-1, indicating a bidirectional interaction between leptin and inflammation [65]. In addition to its interaction with cytokine pathways, leptin stimulates the production of CC-chemokine ligands (CCL3, CCL4, and CCL5) in macrophages [64]. Leptin can also trigger the chemotaxis and ROS production of neutrophils as well as the differentiation, proliferation, activation, and cytotoxicity of natural killer (NK)-cells [66]. Moreover, it inhibits apoptosis and improves the activation and proliferation of T lymphocytes [67]. It also promotes the Th1 phenotype of lymphocytes and the production of IL-2 and IFNγ, while inhibiting the Th2 type and the expression of IL-4 [68].

Another important adipokine is Resistin, belonging to the cysteine-rich family of resistin-like molecules (RELMS) and named according to its connection to the resistance to insulin [69]. The ability to induce insulin resistance is associated with activation of SOCS3, an inhibitor of insulin signaling in adipocytes. However, this has so far only been observed in mice, not in humans [70]. Resistin levels are upregulated in the adipose tissue as well as in the serum of obese individuals [71]. The localization of resistin production seems to differ between mice and men: Whereas the synthesis in humans takes mainly place in macrophages and monocytes [72], it is in mice predominantly produced in adipocytes [73]. Furthermore, the human type only shares a 64% homology with the murine [74]. Nevertheless, it has a known pro-inflammatory effect in humans, since resistin stimulates the expression of TNF and IL-6 in mononuclear cells [75] as well as the expression of pro-inflammatory adhesion molecules vascular cell adhesion protein 1 (VCAM-1), intercellular adhesion molecule 1 (ICAM-1), and pentraxin in endothelial cells. Pentraxin directly counteracts adiponectin, an anti-inflammatory adipokine, and fosters the adhesion of leukocytes [76].

Visfatin, also known as nicotinamide phosphoribosyltransferase (NAMPT), represents another type of adipokine, mainly secreted by the visceral type of adipose tissue [77]. In humans, an elevated level of visfatin has been detected in obese and type 2 diabetes patients [78]. Moreover, a positive correlation between IL-6 and the c-reactive protein (CRP) has been observed, linking visfatin to inflammation [77]. In 1987, the first adipokine, adipsin, was described, also known as complement factor D, a part of the alternative activation pathway of the complement system [79]. It is dysregulated in obesity and diabetes models [80]. In obese mice, circulating adipsin levels are decreased [81], whereas in humans a mild elevation could be observed [82]. In 2014, studying mouse models as well as diabetes patients, Lo et al. [83] could demonstrate that adipsin serves a connection between adipocyte function and ß cell physiology in the pancreas.

In this context, a number of studies have demonstrated that interferons (IFNs) are also released from adipose tissue [84]. Controversially, Surendar et al. showing that adiponectin reduces the IFNɣ level but the hypoleptinemia could be shown as responsible for the decrease of the IFNɣ response [85]. IFNɣ influences the function of adipocytes and promotes the inflammation of the adipose tissue [86]. Studies are showing a shift to the Th1-cytokine profile triggered by IFNɣ [87]. Particularly important is in the context of viral pneumonia that the IFN response is the most efficient innate immune response against viral infections [88] The inhibition of the viral replication mediates the antiviral effect, and primarily type I IFN (IFNα/β) plays a crucial role.

Adipocytes and stromovascular cells of the adipose tissue are addtionially able to produce the well-studied pro-inflammatory cytokine TNF [33]. Levels are clearly increased in the systemic circulation and adipose tissue in obese individuals as well as in models of type 2 diabetes [89]. TNF has also been proposed to play an important role in the development of insulin resistance since it debilitates the important tyrosine phosphorylation of the insulin receptor and its substrate IRS1 in muscle and adipose tissue [90].

Nearly one-third of circulating IL-6 is produced in visceral adipose tissue, where it is secreted mainly by macrophages and adipocytes [91]. The plasma levels again positively correlate with obesity in humans [92]. IL-6 also represents a link to insulin resistance, as it has been shown to suppress metabolic processes stimulated by insulin in hepatocytes, possibly induced by SOCS3 expression [93]. Elevated levels of IL-6 are likely connected to an increase in acute phase response proteins, such as CRP [25]. In addition to the above-mentioned cytokines, IL-18 is produced in adipose tissues and shows increased levels in obese individuals [94]. In rodent models, this overexpression leads to a higher amount of cell adhesion molecules, the infiltration of macrophages, and vascular abnormalities [95].

Adipocytes and macrophages are further able to synthesize the retinol-binding protein 4 (RBP4); its levels are elevated under obese conditions and associated with features of the metabolic syndrome in humans [96,97,98]. Lipocalin 2 and angiopoietin-like protein 2 (ANGTPL2) are expressed in adipose tissue and positively correlated with adiposity, hyperglycemia, insulin resistance, and CRP levels in humans [99,100]. Additionally, the chemokines CCL2, also known as monocyte chemoattractant protein-1 (MCP-1) [101], and C-X-C-motif chemokine 5 (CXCL5) [102] are secreted by the adipose tissue. *Ob/ob* mutant or diet-induced mice (DIO) mice, as well as obese humans, express high levels of MCP-1. In mice, it has already been shown to activate macrophage recruitment and to promote inflammation, glucose intolerance, and insulin insensitivity [103]. Obese and insulin-resistant human individuals also show an increase in CXCL5 levels, a factor that is produced by macrophages within adipose tissue [102].

### 4.2. Anti-Inflammatory Molecules of the Adipose Tissue

The adipokine with the highest serum levels is adiponectin, which is almost exclusively synthesized by adipocytes [69]. After its discovery in 1995, Hu et al. [104] detected for the first time that obese mice and humans show a downregulated expression of adiponectin in adipose tissue. Conversely to this investigation, especially adipocytes in visceral adipose tissue are the main source of this adipocytokine [105]. It shows structural similarities to the complement factor C1q and is also able to form similar complex structures, such as multimers [106]. Adiponectin has a molecular weight of 30 kDa and accounts for approximately 0.01–0.05% of the plasma protein amount [107]. In the circulation, it is present in different forms, as low-, medium- and high-molecular weight (LMW, MMW, HMW) complexes [108]. Most notably, the HMW type is seen to be the most biologically active isoform [109]. Adiponectin exerts its main functions via the adiponectin receptors 1 and 2 (ADIPOR1 and ADIPOR2), which are expressed in various tissues [110]. The central functions of adiponectin are orchestrated via AMPK signaling [111].

Besides the detected downregulation of adiponectin in subjects with an increased body mass, especially the inverse correlation to glucose intolerance and type 2 diabetes is of importance [112]. In this context, adiponectin appears to promote beta-cell function and survival [113]. Furthermore, it increases insulin sensitivity in hepatocytes [114]. Apart from diabetes, low adiponectin levels are also associated with an increased risk for hepatic fibrosis [115] and cancer [116]. It was also observed, that an elevated level of adiponectin is associated with a decreased susceptibility for myocardial infarction in men [117]. Concerning pulmonary impairments, it is positively associated with lung function in healthy adults [118].

With regard to the inflammatory actions, there are different molecular actions known that suggest adiponectin as a potential antagonist of leptin. Clear anti-inflammatory properties through inhibition of IL-6 production, induction of anti-inflammatory cytokines, such as IL-10 or IL-1 receptor antagonist [119], and reduction in ICAM-1 and VCAM-1 [120] were shown. Secondly, adiponectin levels are decreased in obese and type 2 diabetic subjects and are negatively correlated with visceral mass [121]. Furthermore, the expression seems to be downregulated by pro-inflammatory cytokines TNFα and IL-6 [106], hypoxia, and oxidative stress [122]. It also negatively correlates with the level of CRP in obese or diabetic conditions [106]. Adiponectin can also affect macrophages by stimulating the production of anti-inflammatory cytokines [123]. Along the same line, adiponectin-deficient mice display an increased expression of pro-inflammatory M1 type markers and decreased anti-inflammatory M2 type markers [124].

In 2010, another anti-inflammatory adipokine, secreted frizzled-related protein 5 (SFRP5), was discovered [125]. The level of SFRP5 is downregulated in adipose tissue from obese rodents as well as from obese humans with insulin resistance [63]. A deficiency of this adipokine leads to an accumulation of macrophages resulting in increased pro-inflammatory cytokine production [63]. A clinical study furthermore demonstrated an association between lower SFRP5 levels in adults with impaired glucose intolerance and type 2 diabetes, as well as a negative correlation to increased BMI [126].

The overall dysregulation of secreted adipocytokines creates a low-grade chronically inflamed environment during weight gain conditions. This chronic inflammatory state is closely linked to the predisposition to various comorbidities of obesity: lower levels of adiponectin, for example, are known to lead to an elevated risk to develop cardiovascular diseases, such as hypertension [127] or myocardial infarction [117]. This dysregulation is further connected to the development of insulin resistance [128] and potentially impacts the comorbidities of cancer and asthma [31]. Moreover, a connection between this chronic state of low-grade inflammation and the susceptibility to viral or bacterial pulmonary infections seems likely. In Chapter 5, we summarize, connect, and compare different publications on pulmonary infections, the most frequent infection focus, and discuss obesity as a risk factor for severe infections. Different experimental setups to mimic obesity in vivo and in vitro are summarized in Chapter 4 beforehand.

## 5. Experimental Model Systems to Study Molecular Effects of Obesity

### 5.1. In Vivo Models

In the last decades, a variety of rodent models aiming to mimic the human pathophysiology of obesity have been developed. They can be either of monogenic or polygenic origin [129]. The most prominent monogenic obesity model is the *ob/ob* mouse characterized by a global lack of leptin on protein- but not mRNA-level [130,131]. Based on this mutant and parabiosis experiments, leptin was discovered in 1994 [44]. The *ob/ob* mutant displays very prominent obese phenotypes that include induction of metabolic alterations such as insulin resistance and hyperglycemia, and can vary in different mice strains [132]. A second strain used for parabiotic experiments, the *db/db* mouse, shows the same obese phenotype. In this case it is not orchestrated by the absence of leptin protein but by a global lack of the leptin receptor. Since this mutant is further defined by early-onset and severe progression of diabetes, it is more commonly used in the diabetes context than in obesity research per se [133]. In CPE^fat/fat^ mice, an obese phenotype with diabetes is induced by the *fat* mutation in the Carboxypeptidase E (*cpe*) gene, an enzyme important for the synthesis of insulin [134]. Another rodent model is the Zucker rat, discovered by Zucker and Zucker in 1961 [135]. This strain is defined by a homozygous mutation in the gene fatty *(fa/fa)*, leading to a deregulated food intake via defects in the leptin signaling. The Zucker rat mimics the state of hyperleptinemia and further develops insulin resistance but no diabetes [129].

Monogenic models, however, cannot fully resemble the origin of obesity in humans, because only a small proportion of obese individuals actually feature mutations in leptin or its receptor [136]. In contrast, people with an obese phenotype display a leptin resistance, defined by high protein levels in the circulation with no means to respond appropriately [137,138]. The DIO mouse is a prominent, polygenic obesity model, resembling the human pathophysiology much closer. The overall outcome of the diet varies according to the type and composition of food and the rodent strain utilized [129]. Frequently, fodder is used that is predominantly composed of soybean and lard with a kcal fat content of either 45% or 60% [129]. Another polygenic diet-based model is the Cafeteria DIO. Here, more enjoyable food, like candy, is given additionally to normal fodder resulting in a more humanized eating behavior [139]. However, problems in standardization are considerable drawbacks. The DIO concept was further improved in rats by Levin and colleagues in 1989. The group started to mate mice with an equal weight status after the high-fat diet (HFD) period. Following outbreeding, offspring from the obesity-prone group had a higher probability to gain weight during HFD from a very early stage on [140].

Although, rodents can present a suitable model for human obesity in some settings, the effort and costs to provide appropriate husbandry conditions are rather high. Therefore, other in vivo approaches using non-mammalian models are in use: Especially the zebrafish shows a variety of synergies with mammals, in particular regarding adipose tissue and lipid metabolism, making it a promising alternative in vivo model to study obesity [141]. Its high reproduction rate, low husbandry costs, and phenotypical transparency enable large-scale experiments, for instance, in the framework of anti-obesity drug development [142].

### 5.2. In Vitro Models

Even though valuable insights on the molecular effects of obesity could be gained from animal models, there is an urgent need for alternative model systems allowing human-centric research. Despite the increasing humanization of animal models, there still are major physiological discrepancies. Especially with regard to metabolism, functioning of the immune system, and general response strategy to infections, mice differ significantly from humans [143,144,145].

However, establishing a suitable human cell source for in vitro model is a major challenge. Even in the case of ready abundance of adipose tissue as a byproduct of cosmetic surgeries, for instance, culturability aspects including expandability or cryopreservation of cells restrict full flexibility. Alternatively, human cell lines are broadly used; however, these cell lines are mostly derived from cancerous tissues. A very promising approach merging the advantages of both human primary cells and cell lines is the stem cell (SC) technology. Despite the speedy progress and daily achievements in the field, the maturity of SC-derived cells, e.g., adipocytes, is often still insufficient: Adipocytes derived from stem cells or pre-adipocytes in vitro were found to (i) hardly reach a state of unilocularity (i.e., their lipids are stored in multiple smaller vacuoles as compared to the mature adipocyte phenotype of just one large lipid vacuole) and, (ii) to secrete adipose-associated messenger molecules at very different proportions than in vivo [146]. While both phenomena reflect an immature adipocyte state, the latter in particular renders an in vitro differentiation of adipocytes unsuitable when studying molecular patterns of obesity, which is characterized by altered cytokine and hormone profiles [147,148].

Advanced 3D systems, such as organoid technology, often lack key in vivo microenvironment characteristics including vascular perfusion, mechanical cues, tissue-tissue interfaces and recapitulation of immune aspects. A step further towards a physiologically relevant in vitro culture system presents the Organ-on-Chip (OoC) technology. OoCs are microfluidic devices that are capable of emulating human biology in vitro at the smallest possible scale. Alongside three-dimensionality, on-chip microenvironments tailored to specific organs, as well as physiological cell-cell interactions, OoCs enable the application of dynamic fluid flow. Especially in combination with both the iPSC and organoid technologies, OoCs have the potential to lead to a paradigm shift, moving non-clinical research from animal models to advanced human-based in vitro systems [149,150].

Adipocytes are a unique set of cells that are capable of storing large amounts of intracellular lipids without being damaged. However, these unique characteristics of mature adipocytes pose challenges when they are supposed to be cultured in vitro, the most severe being an enormous fragility and size (up to 200 µm cell diameter; almost the entire cell body consists of one large lipid vacuole) as well as buoyancy in aqueous liquids owing to the lipid content. Moreover, mature adipocyte in vitro culture is limited by a non-proliferative and non-cryopreservable status of the cells making donor-specific experiments almost impossible. As a result, it comes as no surprise that many of the existing adipose in vitro models rely on an in vitro differentiation of stem cells or adipose progenitor cells into adipocytes. Pre-adipocyte-derived adipose organoids, for instance, constitute simple but promising 3D models to study adipose inflammation [151,152,153]. Hence, unless studying aspects of adipogenesis, as done in many models discussed in a recent review [146], a conservation of maturity of primary adipocytes in vitro defines the cell source of choice when studying impacts of adipose tissue function.

To date, research in the field of adipose tissue in vitro models is rather sparse in comparison to other organ systems. In addition, the majority of adipose tissue engineering aims at the development of large-scale tissue for use as grafts, rather than focusing on research of adipose tissue (patho-) physiological mechanisms. As discussed above, conventional dish culture of mature adipocytes is not possible owing to buoyancy and fragility issues. One common solution to handle mature adipocytes ex vivo/in vitro is the use of 3D scaffolds from various biomaterials [154,155,156,157]; silk [154,155,158], modified cross-linkable gelatin [156] or collagen [157]. A further approach towards engineering structural support for adipocytes is the so-called sandwiched WAT (SWAT): primary human adipocytes are cultured in between two tissue-engineered sheets from adipose-derived stromal cells [159]. Compared to approaches creating mechanical support, other concepts even take advantage of the mature adipocytes’ buoyancy. Based on adipocyte floating culture, Harms et al. [160] introduced a versatile technique to culture and maintain identity and function of mature adipocytes from various origins (murine vs. human, subcutaneous vs. visceral, and lean vs. obese), dubbed ‘membrane mature adipocyte aggregate cultures’ (MAACs) (Figure 3A).

While the previously discussed 3D in vitro models integrating mature adipocytes present valuable advances in terms of long-term culturability, they still fall short of mimicking physiological adipose tissue microenvironment as well as vasculature-like perfusion. Since comprehensive reviews on adipose tissues-on-chip were published only recently [147,161], we will highlight solely a few examples which exhibit the greatest potential for studying the influence of adipose tissue on lung physiology.

A key aspect to studying molecular patterns of obesity is the preservation of physiological hormone and cytokine secretions in vitro as well as suitable means for assessing these secretions. Thus, for the interrogation of adipocyte functionality, microanalytical fluidic systems (MAS) are of utmost value: One of the first MAS allowed for time-resolved sampling of factors secreted by endocrine tissues, such as adiponectin [162]. More advanced devices based on droplet microfluidics significantly increased the time-resolution of the secretion sampling [163]. Combined with on-chip enzyme assays, the droplet-based sampling approach enabled monitoring of glycerol secretion from adipocytes at a 3.5 s temporal resolution (Figure 3B) [164].

While these microanalytical microfluidic devices have a great potential for assessing highly temporally resolved secretion readout, they are less suited for long-term culture of tissues on-chip and for studying the interaction with other tissue types. The majority of adipose-tissue-on-chip culture systems introduced so far are based on in vitro differentiated adipocytes [146]. Owing to the vast prevalence of type 2 diabetes, studying molecular mechanisms of adipose inflammation in the context of insulin resistance weighs heavy in the field of adipose tissue research. Accordingly, a number of adipose tissue OoCs with a focus on adipose-immune interaction were developed. Zhu et al. [165] co-cultured mouse adipocytes derived from the 3T3L1 precursor cell line and macrophages (J774A.1 cell line) in a microfluidic platform with integrated biosensors for cytokine detection (Figure 3C). Upon injection of stimulating factors, they were able to detect in situ the kinetics of pro-inflammatory (TNF-α and IL-6) and anti-inflammatory (IL-10 and IL-4) cytokines. Liu et al. [166] co-cultured human pre-adipocyte-derived adipocytes and macrophage-like cells (U937 cell line) in two different compartments of a microfluidic platform and measured the cytokine release and glucose uptake. Given the importance of physiological hormone and cytokine secretion profiles from adipose tissue, the integration of mature adipocytes into OoC platforms will be crucial when studying the effects of molecular secretions of adipose tissue on other organs in vitro. An OoC that accommodates mature human adipocytes was recently introduced [167] r: in a microfluidic platform specifically tailored to the requirements for adipocyte culture. The viability and functionality of mature adipocytes isolated from human subcutaneous adipose tissue could be maintained for up to 5 weeks (Figure 3D). Although being able to include mature adipocytes, this system still does not fully recapitulate in vivo adipose tissue since it lacks further cell types such as adipose tissue macrophages and endothelial cells, especially of interest for studying adipose tissue inflammation and obesity.

Besides the recapitulation of the local micro-physiological environment, a further unique aspect of the OoC technology is the capability to connect different tissues with each other, enabled by the vasculature-like perfusion [168]. These multi-organ-chips allow the study of tissue-tissue interactions [169], investigation of pharmacokinetictics of drugs [170] and modeling of systemic diseases such as diabetes [150]. In the context of the impact of obesity on bacterial and viral lung infections, two-organ-chips integrating adipose and lung tissue could potentially path the way for entirely novel types of mechanistic studies with direct human-relevance.

In conclusion, a number of adipose tissue-on-chip and advanced 3D adipose tissue culture models have emerged in the past years. These systems address many of the limitations of animal models and clinical studies by providing human-relevant models amenable for patient-specific experimental studies on a mechanistic level. The majority of the models study inflammation within the scope of insulin resistance/diabetes. However, they also provide a great potential to study how the low-grade inflammation in obesity affects lung infection, especially when connected to lung-on-chip models [171].

## 6. Pulmonary Infections and the Impact of Obesity

Infections in the pulmonary tract can be either of bacterial or viral origin. In the following, we summarize observations and interpretations from clinical and more experimental studies in order to further elucidate obesity as a risk factor in both infection scenarios, as summarized in Figure 4.

### 6.1. Bacterial Lung Infection

Investigating potential correlations between weight, infection risk, and overall mortality in clinical studies is the most straightforward approach to analyze obesity as a potential risk factor for the development of bacterial infections. In the case of community-acquired pneumonia (CAP), mainly caused by bacteria like *Streptococcus (S.) pneumonia* [172], the results are in general following a clear trend; although they are in part contradictory.

Data of various groups are pointing towards the so-called “obesity paradox”. This paradox describes the observation that an obese weight status can be protective in some health concerns, which is also observed in case of non-infectious conditions such as heart disease [11] or end-stage renal disease [173]. Supporting this hypothesis for CAP, Corrals-Medina et al. [174] could show that an increased BMI is associated with reduced 30-day mortality; confirmed in a larger sample size via a clinical study by Singanayamagam et al. [175]. Similar results were reported for 1-year [176] and 6-year mortality rates [177]. Interestingly, one study also reported higher CRP levels in obese subjects [175]. This suggests higher levels of inflammation, which were assumed to possibly improve outcomes via immunomodulation in severe cases. In stark contrast, Chen et al. [176] did not observe this effect, but determined CRP levels as one of the seven risk factors besides obesity for increased 1-year mortality and suggested a potential association between those parameters. Singanayamagam et al. [175] further argued that the state of chronic inflammation in obese patients could be a reason for a generally activated state of host defense. Even though, there are some limitations to this study (e.g., potentially misdiagnosed CAP in obese patients), these very recent findings seem to support the concept of the obesity paradox during CAP.

Contradictory to that, a prospective study showed a direct association between excessive weight gain and a nearly twofold higher risk of CAP in women in the US [178]. The limitation of this study is the lack of etiological analysis. The causative pathogen was not identified; thus, it is unclear whether viral or bacterial pneumonia was the trigger. In addition, previous antibiotic treatment or vaccination was disregarded.

To unravel the possible connections between obesity and the risk of bacterial pneumonia/lung infection or associated mortality, further studies tried to examine the underlying inflammatory profiles in various in vivo models infected with different pathogens.

*Ob/ob* mice infected either with *S. pneumonia* [179] or *Klebsiella (K.) pneumonia* [180] displayed higher lethality or mortality and reduced bacterial clearance in lung and blood in comparison to lean mice. While the number of leukocytes and the level of TNF-α, IL-12, and CXCL-2 cytokines were not altered two days after infecting *ob/ob* mice with *K. pneumonia*, alveolar macrophage phagocytosis and leukotriene B4 synthesis were impaired [180]. Infection of *ob/ob* mice with *S. pneumonia*, however, caused increased levels of TNF-α, macrophage inflammatory protein 2 (MIP-2), and prostaglandin E2 (PGE2) as well as a greater number of leucocytes after two days [179]. When analyzing the impact of obesity on *S. pneumonia* infection with the CPE^fat/fat^ mouse model [181], neither differences in survival, in bacterial burden, nor in lung or serum cytokines could be observed [181]. Although some of these findings indicate a link between bacterial infections and obese states, the observations differ depending on the type of bacteria and the model used for the obese phenotype.

The importance of the employed model was highlighted in a study by Ubags et al. [182] that infected four common mouse models (*ob/ob, db/db*, CPE^fat/fat^, and DIO) with *K. pneumonia*. While all showed impaired clearance and neutrophil chemotaxis, they also exhibited differences in granulocyte-colony stimulating factor (G-CSF) mediated survival, cytokine transcription, MAPK, and STAT3 activation. Moreover, temporal variations of the host defense between the groups were observed, which was interpreted as either different bacterial handling by the four models in an early and a late phase or the “plateau” phenomenon. This phenomenon implicates that the bacterial burden of the lung cannot rise above a particular degree in obese mice. Thus, this study showed an impaired pulmonary host defense after bacterial infection, mainly associated with impaired cytokine production or chemotaxis of neutrophils, but also highlighted the variations between the mouse models. Considering all the metabolic discrepancies between the examined models, results from studies using different models should be interpreted with care.

*Escherichia (E.) coli* is another important pathogen rather causing the nosocomial type of pneumonia [183] The impact of obesity on this kind of infection has also been studied with the help of the DIO mice model [184,185,186]. Before the initial infection after the high-fat diet, similarly increased levels of triglycerides, cholesterol, leptin, or IL-6 have been detected [184,185]. In a study by Wan et al. [161,185], mild and severe infections were modeled using two different dosages of the pathogen. The higher pathogen doses resulted in a higher mortality and lung injury in DIO mice, but the difference in *E. coli* counts compared to controls was not observed. Inoculation with lower CFU led to reduced *E. coli* burdens, less severe lung injury, but greater numbers of immune cells and lower IL-10 and TNF-αconcentration in DIO mice compared to lean type. These observations support the hypothesis that during mild infections, obesity is protective, comparable to trends observed regarding the obesity paradox in CAP, but in more severe conditions it might weigh down the host defense. Wang et al. [184] observed a temporal shift in inflammatory signals peaking at 12 h post-infection with *E. coli* in the lean and at 72 h in the DIO mice cohort. This temporal delay was also detected for apoptotic markers within pulmonary cells in the same experimental setup [186].

When interpreting observations in the *ob/ob* mouse model, it is important to keep in mind that they have an obese phenotype caused by genetic ablation of leptin production. This lack of leptin contrasts the high levels in the circulation and their positive correlation with adipose tissue mass in obese humans [129]. Nevertheless, the *ob/ob* mouse model helps to analyze the function of leptin in reactions to infection of the lung. For example, Hsu [179] and Mancuso et al. [180] were able to show that a lack of leptin impairs different aspects of the host defense and that the administration of exogenous leptin was sufficient to restore this defect during infection with *S. pneumonia* and *K. pneumonia*. Even though these results underline the role of leptin during bacterial lung infection, they do not represent a clear link between obesity and lung infection. However, these findings could be of clinical relevance as a potential treatment option for patients with bacterial pneumonia, especially in immunocompromised individuals.

Ubags et al. [187] provided a different hypothesis of the role of leptin for pulmonary host defense: They suggested that rather a hyperleptinemia than high BMI might serve as a link to an impaired pulmonary pathogen defense and increased susceptibility to infection. In two patient cohorts, they could observe a positive correlation between plasma leptin levels and the risk of respiratory infection and further with mortality in patients with severe pneumonia resulting in ARDS. They substantiated these findings by showing a stronger association between leptin levels and lung bacterial burden in DIO mice challenged with *K. pneumonia* than with their body weight. Furthermore, lavage neutrophils of obese and lean mice showed a strong inverse association between neutrophil counts and plasma levels of leptin. To finally separate hyperleptinemia from obesity, they created a hyperleptinemic but lean mouse model. Isolated hyperleptinemia resulted in an increase in lung bacterial colony-forming units (CFU), a decrease in bronchoalveolar lavage (BAL) neutrophil count, and impaired function. In summary, these results indicate that the adipocytokine leptin itself might alter the pulmonary host defenses resulting in increased susceptibility to and mortality from bacterial respiratory infections in obese patients.

To date, no unifying concept can be established based on the summarized findings. The “obesity paradox” seems to hold true for most of the clinical studies performed but the underlying molecular mechanisms still remain unclear. However, it should be considered that the investigated protective role is solely based on correlations to the BMI, a very simple classification of obesity (see chapter 2). In this context, Despres [188] concluded, that the obesity paradox should be rather seen as a BMI paradox. This could also hold true for the protective role in CAP, as especially the body fat composition and distribution change the inflammatory fate of adipose tissue. Attempts to elucidate the mechanisms resulted in contradictory findings. These contradictions, however, are clearly connected to the choice of in vivo model and the pathogen used for the lung infection. Regarding the potential impact caused by the dysregulated adipocytokine network, solely the impact of leptin could be elucidated utilizing *ob/ob* mice. However, there is still the need to further analyze these molecular interactions.

### 6.2. Viral Induced Lung Infection

The majority of respiratory infections are caused by viruses. In particular, the RNA viruses influenza and coronavirus have a large socio-economic impact and are able to cause serious infection courses.

The annual influenza season is a constant in our health care system. It can result in either a mild infection or, in the worst case, severe viral pneumonia. This has led both to the development of diagnostic methods and annual vaccination options as well as to a comprehensive understanding of the infection process of this special type of human pathogen. The emergence of a novel H1N1 IAV strain in California and Mexico in 2009 resulted in the first pandemic influenza period of the 21st century [189]. The virus spread rapidly around the world, leading to 74 countries and territories confirming infections in June 2009 according to the WHO. Genome analyzes of the virus revealed a non-human origin and a relation to known viruses circulating in pigs. The North American H3N2 triple-reassortant, the classical H1N1, and the Eurasian ‘avian-like’ swine H1N1 viruses comprise the origins of the new subtype, able to infect humans [190,191]. The analyzes of the epidemiological profile of infections with this new subtype allowed new insights like the identification of new potential risk factors. At first, these data showed a change in the age distribution, as in the US 60% of the patients were not older than 18 years and only 5% of the infections occurred in patients older than 50. Besides other well-characterized risk factors, obesity appeared as an additional risk factor for the first time [192].

In an extensive meta-analysis, Fezeu et al. [193] included articles from the USA, Ireland, France, and the Netherlands, all aiming to analyze the connection between obesity and influenza A virus (IAV) infection. By analyzing correlations between confirmed IAV-infection, hospitalization, admission to ICU, and body weight status of the patients, they concluded an elevated risk for ICU admission and overall mortality for extremely obese patients (BMI higher than 40). In addition, patients with a BMI higher than 30 showed a moderately but not significantly increased risk for ICU admission and death [193]. In line with these findings, another case-cohort study compared data from the pandemic series of 2009 with another cohort from 2003 to 2006 and confirmed the link between particularly strongly obese individuals and hospitalization and ICU admission [194]. The results from the examination of Braun et al. [195] highlight the differences between the pandemic in 2009 and seasonal IAV. For seasonal IAV, no correlation between obesity and admission to ICU or artificial ventilation could be detected; instead, a rather decreased risk for the development of pneumonia caused by seasonal IAV infection was connected with obesity.

This apparent strain dependency was substantiated in an experimental in vivo study infecting DIO mice either with a pandemic strain from 2009, a seasonal H1N1 strain, or a highly pathogenic Sw31. Increased mortality in obese mice compared to the corresponding lean control group was solely seen for infections with the pandemic strain [196]. Other groups confirmed the increased mortality in response to infection with a pandemic IAV strain in DIO mice [197,198]. Similar results were obtained with *db/db* mice [199]. Further experimental steps addressed the elucidation of molecular events underlying the observed differences in survival, e.g., by monitoring pro-inflammatory cytokines present in the serum of infected obese and lean mice. In the DIO mouse, infection with IAV led very early to a significant elevation of systemic pro-inflammatory cytokines, like IL-1β, TNF-α, IFN-γ, and IL-6, in obese mice compared to non-obese mice [196,197].

Investigation of local molecular alterations within the pulmonary tract showed a significant elevation of active viral particles at day 4 p. i. in mice with DIO [197]. The same effect could be observed in the *db/db* mouse model [199]. However, several other studies found no evidence for alteration of active viral particles within the lung with monitoring starting from day 2 until day 9 p.i. [196,198,200].

Regarding the inflammation profile during the infection, Zhang et al. [197] and Milner et al. [198] could clearly show a heightened local inflammatory state within lung tissue, e.g., by the elevation of the cytokines IL-6 at 4 days p.i. and further upregulation of the chemokines MCP-1 and KC at day 8 within the infected lung of obese mice These results are contrasted by several studies reporting lowered levels of cytokines [196,199,200]. In particular, Smith et al. [200] detected decreased amounts of IL-6, TNFα, IL-1β, and IL-10 in obese mice’s lungs as well as lowered levels of the chemokines MCP-1 and RANTES. These observations, i.e., the downregulation of pro-inflammatory cytokines, are discussed as a potential explanation for the reduced viral clearance in obese mice. One potent contributor to the lowered inflammatory milieu might be a decreased infiltration with macrophages, the main source of pro-inflammatory cytokines during lung infection [201]. This suggestion is in accordance with the reduced amount of monocytes differentiating into macrophages detected in obese patients compared to normal-weight individuals [202].

Regarding systemic leptin levels during infection with IAV, findings are quite variable, including both reports of upregulation of leptin in lean mice only and of constant leptin levels during infection [196,197,200]. Elevated amounts in the circulation could, in principle, influence the inflammatory processes and viral spreading within the pulmonary tract. Elevated levels of leptin in the pulmonary tract could also result in an upregulation of its downstream signaling branches, e.g., the negative regulator SOCS3. SOCS proteins are also activated by different cytokines such as IL-6 and IL-4 via JAK/STAT and play an important role in immunity [203]. The ability to likewise inhibit the signaling of these cytokines highlights the cross-regulatory role of the signal mediator SOCS [204]. Moreover, it has been shown that SOCS1 and SOCS3 suppress the type I IFN response, a central defense system against viral infections [205], e.g., in case of IAV infection [206]. According to these findings, Smith et al. were also able to detect a downregulation of IFNα and IFNβ in the lungs of obese mice after infection with PR8. In this context, Lui et al. [207] determined a reduction in virus titer in a corresponding SOCS3-knock-out mouse strain. Thus, the upregulation of SOCS3 as one explanation for the leptin resistance could lead to the downregulation of pro-inflammatory signaling processes and via that support the viral replication within the lung.

However, there is also evidence supporting a differing role of leptin signaling during infection of obese mice’s lungs. As mentioned above, mice with a global loss of the leptin receptor showed reduced viral clearance. The same study, however, detected improved survival in mice that were solely leptin receptor-deficient in lung epithelial cells, alveolar type II cells, and monocytes suggesting, that leptin signaling in these cells enables a higher number of viral particles [199]. Moreover, treatment of obese mice with anti-leptin-antibodies resulted in improved survival and downregulated cytokine levels, emphasizing again the role of leptin in obese mice during the course of infection. Nevertheless, leptin administration did not result in enhanced viral clearance [197].

A study from Paich et al. [208] addressed another important aspect, the impact of obesity on immune cells of patients in response to pandemic H1N1 (pH1N1). With the help of the ex vivo infection of peripheral blood mononuclear cells (PBMCs) from patients with different body weight status, they observed a dysregulation and lowered activity of CD4+ and CD8+ T cells in obese and overweight. These results are in line with the inefficient response to vaccination in obese individuals, already described in a prospective observational study [209]. To elucidate this connection, several studies were performed in mouse models. DIO mice that were previously immunized with commercial monovalent 2009 H1N1 vaccine showed a lowered antibody response and neutralizing capacity, resulting in elevated viral titers, more severe lung pathology, and inflammatory response after infection with H1N1 [210]. Karlsson et al. [208] reasoned that this could be due to an impaired memory response in obese mice, since molecular alterations were detected in response to a secondary infection with an H1N1 strain after a previous H3N2 infection. Furthermore, interferences within the antigen-presenting dendritic cells (DC) as important initiators of T cells were described, leading to alterations in the number and functionality of CD3 + and CD8+ T cells in lung tissue [211]. Kosaraju et al. [212] additionally determined disturbances in B cell activity both in obese humans and in obese mice.

In conclusion, results from both, clinical studies and in vivo obesity models, highlight the increased susceptibility for an IAV infection and decreased survival associated with obesity. Surprisingly, also a strain dependency of the effect, particularly for the pandemic strain of 2009, was discernible. Obese patients are not only at higher risk for an infection per se but also the lowered effectiveness of vaccination is an important issue for our health care system. So far, no clear concept of the molecular interplay between physiological effects of obesity and IAV infections could be established.

Besides an annual influenza season and recurring pandemics, we are currently experiencing a new coronavirus-triggered pandemic, with a novel coronavirus (SARS-CoV-2) causing the coronavirus disease (COVID-19) and challenging clinical management worldwide. Continuously rising numbers of clinical studies are being published worldwide that indicate risk factors for a severe course. These factors include pre-existing lung disease, neoplasms, age, but also overweight and obesity [213,214,215]. Clinical observations identified obesity as an independent risk factor for hospitalization, ICU treatment, and death [9,10]. Large cohort studies describe significantly elevated risks of intubation and death for obese COVID-19 patients [216,217,218]. Interestingly, younger obese individuals are particularly affected [219,220].

Besides these clinical observations, Huizinga et al. [221] hypothesize that the inflammatory response of the obese tissue leads to a dysregulated immune response. Due to SARS-CoV-2 infections, cytokine storms are frequently reported [222]. Due to a large number of monocytes and macrophages in the fatty tissue, there is an increased activation, which further exacerbates the inflammation. In addition, the authors assume an impact on viral clearance caused by obesity. However, the detailed immune mechanisms are currently insufficiently clarified. Krams et al. [223] strongly emphasize the need to differentiate between visceral and subcutaneous adipose tissue. The authors highlight that primarily the visceral fat is linked to severe COVID-19 course of infection [224,225], as this tissue has an increased endocrinological activity [226].

A further essential aspect is the production of the transmembrane protein angiotensin-converting enzyme 2 (ACE2) receptors by the adipose tissue [227,228]. SARS-CoV-2 begins its replication cycle with the entry process into the host cell. For this, the binding of the spike glycoprotein to ACE2 is a crucial step [218]. Besides the lung, various tissues and organs express ACE2, and viral RNA has already been demonstrated in many extrapulmonary tissues [229]. Besides the small intestine and kidney, expression levels of ACE2 are particularly high in adipose tissue [230]. On this basis, adipose tissue could be actively considered as a replication site for the virus and thus become a source of viremic virus particles.

## 7. Conclusions

In this review, we summarized the role of obesity as a risk factor for pulmonary infections either of viral or bacterial origin. The most obvious effects stem from mechanical complications in the pulmonary tract impacting the susceptibility for infections. On a molecular level, the endocrine functionality of adipose tissue further influences pulmonary function and infections. We particularly focused on the dysregulation of these adipocytokines in the remodeling process during weight gain.

Particularly data gained from clinical studies highlight the impact of obesity on the course of respiratory infections. In the case of CAP, triggered by bacteria such as *S. pneumonia*, the results mainly point towards the well-described “obesity paradox”, suggesting that high body weight is protective. For IAV infections, however, obesity was defined as an independent risk factor for severe courses following the pandemic of 2009. These contradictory findings highlight the differences between pulmonary pathogens and display the complexity of risk factor assessment. However, no unambiguous trend concerning the course of respiratory infections could be reasoned, neither regarding infections of bacterial nor of viral origin. With regards to the molecular changes resulting in a state of low-grade inflammation in obese patients, it is unclear whether this state is a benefit or a disadvantage in this setting. However, both pro- and anti-inflammatory stimuli influence the host and thus the course of a respiratory infection. These effects are variable in animal models based on the pathogen, the concentrations, but also the model used.

To study the impact of obesity, a variety of in vitro or in vivo model systems are used. Particularly rodents can mimic an obese weight status and associated physiological, pathological and molecular alterations. However, the translation to humans is often difficult and interpretations of the observations are dependent on the respective origin. Monogenic rodent models such as the prominent *ob/ob* mice indeed feature a clear obese phenotype; polygenic more diet-based systems, however, seem to resemble the situation in obese humans better. Besides rodents, other model organisms such as zebrafish and drosophila are also utilized; So far, however, rather in the context of drug development than of infectious diseases. An alternative to animal models are in vitro models. Conventionally, 2D cell culture assays relying on (murine) cell lines or primary cells have been utilized. In recent years, technological advances such as iPS-cells, 3D-cell culture, and organ-on-chip have led to a novel generation of in vitro models that open up novel possibilities to study obesity and its impact on pulmonary infections.

It is well known that pulmonary infections are based on complex molecular interactions and immune reactions. Obesity influences a variety of physiological processes, including the dysregulation of signaling molecules, such as leptin. Hence, the elucidation of a clear molecular concept detailing the multistep processes of obesity and lung infections and their interactions is extremely complex. However, the ability of respiratory pathogens to cause pandemic episodes, as for IAV in 2009 and the current COVID-19 pandemic, emphasizes the need for further mechanistic investigation of risk factors. The fact, that obesity on its own is a health problem of already pandemic measures in our century, additionally highlights the clinical relevance of the issues outlined in this review.

## Figures and Tables

**Figure 1 ijms-22-03456-f001:**
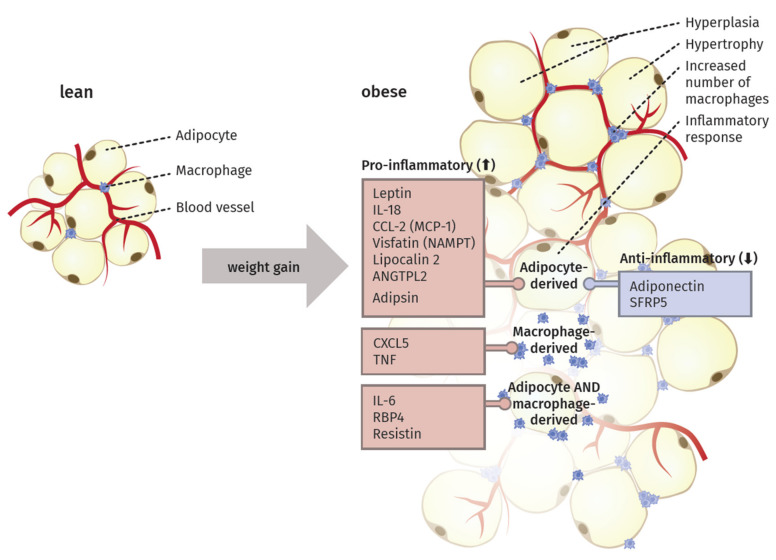
The remodeling process in adipose tissue during weight gain. White adipose tissue is primarily composed of adipocytes, macrophages and blood vessels. To cope with an elevated amount of triacylglycerols (TG) during weight gain, adipocytes increase in size (hyperplasia) and number (hypertrophy). This can further result in the apoptosis of adipocytes, chemotactic regulations, hypoxia and free fatty acids. All four mechanisms are discussed to foster the heightened migration of macrophages into the adipose tissue. Moreover, the secretions of different adipocytokines, produced in adipocytes, macrophages or in both cell type, are influenced. In general, anti-inflammatory secretory products are downregulated, whereas the secretion of pro-inflammatory adipocytokines is elevated. These developments provide a low-grad pro-inflammatory environment, which arises from the increased and remodeled adipose tissue in obese subjects.

**Figure 2 ijms-22-03456-f002:**
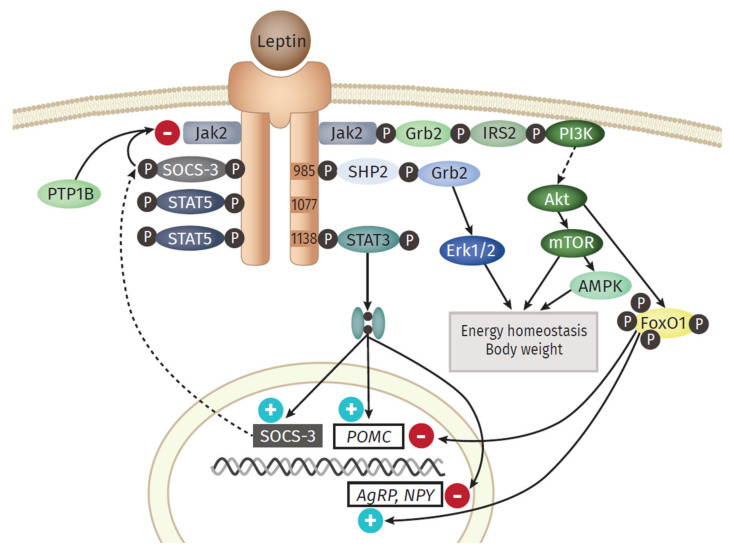
Overview of the leptin signaling pathway. The long isoform of the leptin receptor (ObRb) possesses the whole intracellular domain, with the conserved tyrosine residues (Y985, Y1077, Y118). The central function of leptin in the hypothalamus is regulated via the activation of anorexigenic neurons and orexigenic Neuropeptide Y (NPY)/Agouti-related Protein (AgRP) neurons. The most prominent downstream signaling component of the leptin receptor is the Janus kinase (JAK)/signal transducer and activator of transcription (STAT) pathway. After binding of leptin, subsequent dimerization of the receptor and the following activation of JAK2 occurs, followed by the recruitment of the Src homology 2 (SH2) domain of STAT3 to the conserved phosphorylated tyrosine Y705 residue. Hereinafter, STAT3 itself is phosphorylated by the JAKs at position Y705, dimerizes, and translocates into the nucleus. At this point, it serves as a regulator of the expression of different STAT3 responsive genes. One prominent downstream gene is the suppressor of cytokine signaling 3 (SOCS3), which itself acts as a potent negative regulator of the leptin signaling, by binding of Y985 domain and following inhibition of JAK2. Protein tyrosine phosphatase 1 B (PTP1B), produced in the endoplasmic reticulum (ER), is further able to inhibit leptin signaling by dephosphorylating JAK2. Besides STAT3, STAT5 is also known to be activated and phosphorylated in vivo. Another downstream signaling branch of leptin is the Mitogen-activated protein kinase (MAPK) pathway. Here, the SH2 domain of the phosphatase SH2- containing protein tyrosine phosphatase 2 (SHP2) binds to pY985, becomes phosphorylated by JAKs, which finally activates MAPK extracellular signal-related kinase (ERK1/2) via recruitment of growth factor receptor-bound protein 2 (Grb2). The phosphatidylinositol 3 kinase (PI3K) signaling pathway is activated by leptin. IRS 2 (insulin receptor substrate 2) binds to ObRb through the SH2B1 domain, able to interact and upregulate JAK2. Those IRS proteins are then capable of binding and activating PI3K yielding the subsequent accumulation of phosphatidylinositol 3,4,5-triphosphate (PIP_3_) and activation of 3-phosphoinositide- dependent protein kinase (PDK1) and Akt. The latter inhibits forkhead box O1 (Foxo1), the mediator of the previously mentioned function of leptin in the hypothalamus. Another component downstream of Akt is the Ser/Thr kinase mTOR (mammalian target of rapamycin), another sensor of the availability of nutrients and stimulator of cell growth, protein biosynthesis, and proliferation. The AMPK (adenosine monophosphate-activated protein kinase) pathway is stimulated by leptin as well, but the outcome differs between the tissues. While activated in hepatocytes and muscle, AMPK is inhibited in the hypothalamus, resulting in inhibition of food intake.

**Figure 3 ijms-22-03456-f003:**
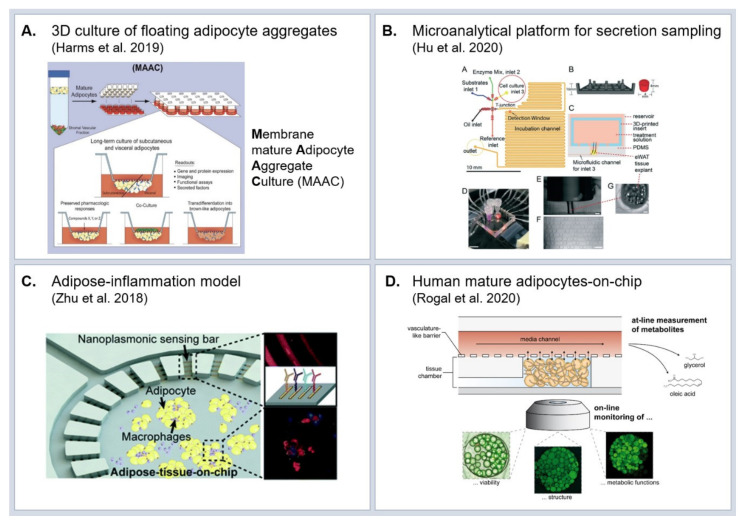
Adipose tissue in vitro models (**A**) Membrane mature adipocyte aggregate cultures (MAACs) enabling long-term culture of buoyant primary adipocytes. Reproduced and slightly modified from [160] (CC BY 4.0); (**B**) Microanalytical fluidic system for the monitoring of glycerol secretion from adipocytes at a 3.5 s temporal resolution. Reproduced from Ref. [164] with permission from The Royal Society of Chemistry; (**C**) Organ-on-chip for studying adipocyte-immune interaction with integrated biosensors for cytokine measurement. Reproduced from Ref. [165] with permission from The Royal Society of Chemistry; (**D**) Adipose-on-chip integrating human mature adipocytes into a micro-physiological environment featuring vasculature-like perfusion. Reproduced from reference [167] (CC BY 4.0)”.

**Figure 4 ijms-22-03456-f004:**
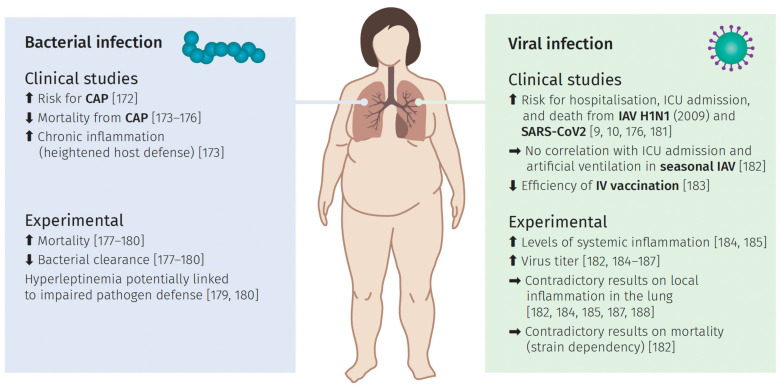
The impact of obesity and bacterial and viral infections. Overview of observations gained from clinical studies and experimental approaches regarding bacterial and viral infections in obese subjects with corresponding sources.

## Data Availability

Not applicable.

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
