# Peer review of "The Inflammatory Profile of Obesity and the Role on Pulmonary Bacterial and Viral Infections"

_ijms, 2021, doi:10.3390/ijms22073456_

Round 1
Reviewer 1 Report
The review article titled “The role of obesity on pulmonary bacterial and viral infections” by Hornung et al., attempts to summarize the impact of obesity in the context of bacterial and viral infections. This review is interesting and important considering the relevance of obesity in pathology of infectious diseases. The article is well-written and cites the relevant literature. Despite the fact that many groups have now shown that there is a link between microbial infections and obesity, a clear cause-effect relationship has not been reached in this field and this article provides a good review of the existing literature on this topic, while maintaining a balanced standpoint in interpreting results/findings from other groups. Although the article title suggests that the review would focus on the part about bacterial and viral infections, a major chunk of the content is just focused on documenting the general aspects of obesity, model systems and a summary of the mechanism behind role of adipocytes in maintaining an inflammatory milieu. Its only towards the end of the review that the authors discuss about bacterial and viral infections and implications for obesity in that. Therefore, I suggest that the title be altered to better reflect the bulk of the content that the manuscript focuses on. Moreover, the authors should discuss the role of interferons in obesity because they play an important role in immune response against bacterial and viral infections as well as in obesity.
Reviewer 2 Report
The article is a very interesting and comprehensive review concerning obesity and bacterial and viral infections. I recommend the article to be published after minor revision done.
- Grammar mistakes and typos should be corrected throughout the manuscript.
- Line 76: It should be: Sharp et al [14] .... - I mean, a citation directly after the authors names - it should be corrected throughout the manuscript.
- Line 82: the term Work of Breathing should be described in some details.
- Line 109: beige and pink types of adipocytes should also been mentioned. It is also worth mentioning that brown adipocytes have been found to be present in adults.
- Chapter 4.2 - In my opinion, this chapter is written in an overcomplicated language. The sentences are too long. I recommend rewriting this chapter so that the sentences are shorter and more understandable.
- Tables summarizing the described studies could be a beneficial addition to the manuscript.
Reviewer 3 Report
Hornung et al. present a review of the current literature on the role of obesity on pulmonary bacterial and viral infections. In the last years – to my knowledge- no similar review was published, and the review contains all significant and important papers in this field. The paper is engaging, informative and of educational values.
I enjoyed reading the manuscript. I commend the authors for several strengths of their work, including addressing an exciting and timely question. The subject is in the range of the journal, and the manuscript is of clinical relevance.
However, I have some suggestions to improve the review.
- The authors should be clear about what they mean by obesity. Many authors have pointed out that reliance on BMI as a sole marker of obesity seems to be the serious limitation of studies on the relationship between obesity and other diseases. They indicated a poor linear relationship between BMI and total body fat and suggested that body fat distribution would be more clinically significant than overall obesity. I suppose visceral obesity may be more critical In the case of bacterial and viral infections as it is a major source of pro-inflammatory adipokines.
- Likewise, this so-called obesity paradox tends to disappear in many diseases if we consider visceral obesity and not BMI or total fat content. I am curious about the authors' opinion on this subject, for example, in the case of CAP.
- Line 139: Sun et al. - there is no reference to the bibliography – probably 35
- Line 297: pro-inflammatory, not pro-inflamamtory
